# The Leaky Feeder, a Reliable Medium for Vehicle to Infrastructure Communications

**Hasan Farahneh** [1,*] **and Xavier Fernando** [2] 

1   Department of Electrical Engineering, The University of Jordan, Amman 11242, Jordan
2   Department of Electrical and Computer Engineering, Ryerson University, Toronto, ON M5B2K3, Canada; fernando@ee.ryerson.ca
*   Correspondence: h.farahneh@ju.edu.jo; Tel.: +962-791-561-196

**Abstract:** Reliable vehicular communications is fast becoming a necessity. Vehicle to infrastructure (V2I) communication, which is critical for safety, is often interrupted when vehicles travel in tunnels. Leaky Feeder (LF) or radiating cable have been the primary solution to provide wireless access in tunnels and mines, but being overlooked until now. The LF is a natural multi antenna transceiver ideal for broadband short rage access. In this work, we model the LF as a linear antenna array and derive the average bit error rate (BER) in Rayleigh fading channel considering Quadrature Phase Shift Keying (QPSK) and M-Array Quadrature Amplitude (M-QAM) Modulations. We consider maximal ratio transmission (MRT) at the transmission end and coherent detection and maximal ratio combining (MRC) at the receiving end. Analytical expressions are derived for the BER. The effects of slot spacing and carrier frequency on the BER are also studied. Numerical evaluations show that the LF is a strong candidate for tunnels with much lower BER than a single antenna transmitter with the same SNR.

**Keywords:** leaky feeder; BER; QPSK; antenna array; Rayleigh fading; channel gain

## 1. Introduction

Vehicle-to-infrastructure (V2I) communications has become a game changer. This not only ensures safety and connectivity but is also critical for fast emerging self driving automobiles. Although usually connectivity with roadside antennas is not an issue, this is not the case when the vehicle travels in a tunnel. A surprisingly large number of people frequently commute using tunnels. For example, just the Beijing subway alone carries about 3.2 billion riders per year. This number is much higher worldwide. The 50 km undersea English Channel tunnel ('chunnel') carries nearly nine million passengers per year. These confined spaces unfortunately report a 4–5 times higher rate of accidents than that in open terrain, and the consequences are usually fatal. For example, after a fierce fire broke out in the Gotthard tunnel, Switzerland, 40 vehicles were fused to a molten metal mass and 128 persons perished. Hence, the importance of achieving reliable V2I communication in dangerous confined spaces can't be over stressed.

Connected vehicles will be the backbone for context awareness and cooperative driving, leading to intelligent transport systems. While IEEE 802.11p has been specified for vehicle to vehicle (V2V) and vehicles to roadside infrastructure (V2I) data exchange, and (3GPP) specifications support vehicle to everything (V2X) services and upcoming 5G standards promise better V2V connectivity in terms of reliability, latency and spectrum efficiency [1–3].

There have been several standardization activities pointing to the offer of consistent communications with low latency and cost. The most famous activity is a Dedicated Short-Range Communication (DSRC). DSRC has been developed using IEEE 802.11p, which is based on Carrier

Sense Multiple Access (CSMA). However, 802.11p suffers from the main limitations related to 802.xx standards, such as poor scalability to high traffic density and poor support of high mobility [4]. In addition, the other activity is built on the LTE-V2V standard. LTE-V2V was developed on the basis of cellular technology and helps a vehicle to communicate with different types of devices, such as pedestrians with a mobile phone, the infrastructure/network, and other vehicles. It focuses deeply on V2V communication to exchange short messages that are used in safety-related applications [5].

Radiating cables, also referred to as leaky feeders (LF), are a leading candidate for providing wireless access in mines and tunnels [6] for years. LF is also used for underground wireless communications in mass transit railways in some places. Already, few Gb/s bit rates using LF are reported [7]. However, the full potential of LF for V2I communications is not fully exploited, which is the focus of this paper.

The LF is essentially a special coaxial cable with carefully designed slots along its length that enable radiating and receiving radio signals. The radiating cable has multiple functionalities: it not only acts as an array of bidirectional antennas but also transfers the DC power required to power up amplifiers and active nodes in the downstream. The LF system is typically less noisy and has enough bandwidth to support multiple radio frequency (RF) signals. The slot spacing, shape, and size determines the loss, radiation patterns, and coupling properties [8].

Although most studies consider the LF as a single radiating element, the LF must be considered as a linear antenna array for accurate analysis [9]. This will pave the way to model an LF system as a multiple-input-single-output (MISO) wireless communication system [10] or a MIMO system if the receiver also has multiple antennas.

The unique aspect of the LF is the *phase shift* among the RF signals emanating from each slot; this is a function of slot separation ($D$) and the carrier frequency ($f$). These RF signals undergo multipath fading in the wireless channel to reach the receiver where they can be combined at the wireless receiver in multiple ways. Several diversity combining schemes have been previously reported. The authors of [11,12] provide a detailed analysis of maximal ratio combining (MRC) under Rayleigh and Ricean fading channels. To the best of our knowledge, there is no previous research that addresses the BER estimation of communication systems that use the LF transmitter.

In this paper, we model a channel by using an LF to be a Reliable Medium for vehicle to infrastructure communications in the confined places like the tunnels or the underground mines, and we derive a generalized closed form expression for the BER of the LF communication system under multipath Rayleigh fading channel and compare it with the BER of a single antenna system with similar parameters. The effects of slot spacing and carrier frequency are also studied. The organization of the paper is as follows. In Section 2, the characteristics of fading channels are presented. In Section 3, the mathematical expression for the BER of the LF communication system under Rayleigh fading channel is presented. Section 4 shows the simulation and numerical results. Finally, Section 5 provides a brief conclusion and directions for future studies.

## 2. Modeling the Flat Fading Channel

In the realms wireless communication systems, the fundamental performance measure is the average signal-to-noise ratio (SNR), which is the statistical average over the probability distribution of the fading. In simple mathematical terms, if $\gamma$ denotes the instantaneous SNR, which is a random variable (RV) at the receiver output because of the effect of fading, then $\overline{\gamma}$ is the average SNR, where $f_\gamma(\gamma)$ denotes the probability density function (PDF) of $\gamma$ [13].

To begin with, let us first define the Moment Generating Function (MGF) associated with $\gamma$ as

$$M_\gamma(s) = \int_0^\infty f_\gamma(\gamma) \, e^{s\gamma} \, \mathrm{d}\gamma. \tag{1}$$

Let us evaluate the average bit error rate (BER) of this system using a low complex but accurate method. Note that the BER depends on random fading, and it is a nonlinear function of the SNR.

The nature of the nonlinearity is a function of the modulation/detection scheme employed [13]. For example, in the multichannel case, the average of the conditional BER over the fading statistics is not a simple average of the per channel performance measure as in average SNR.

The average BER in the case of perfect phase matching between the transmitter and the receiver can be written as

$$P_b\left(E\right) = \int_0^\infty f_\gamma\left(\gamma\right) P_b\left(E \setminus \gamma\right) \mathrm{d}\gamma, \tag{2}$$

where $P_b\left(E \setminus \gamma\right)$ is the conditional probability and may have different solutions depending on the channel fading type and the modulation method. When fading affects a narrow band system (flat), the received carrier amplitude is modulated by the fading gain $\alpha$, where $\alpha$ is an RV with a mean-square value $\Omega = \overline{\alpha^2}$. Equivalently, the received instantaneous signal *power* is modulated by $\alpha^2$. Note that $\alpha$ has a PDF $f_\alpha\left(\alpha\right)$ that depends on the nature of the radio propagation environment.

The signal is also affected by additive white Gaussian noise (AWGN), which is typically statistically independent of the fading amplitude. The noise is characterized by single-sided power spectral density of $N_0$ (W/Hz). Therefore, we define the instantaneous SNR per symbol by $\gamma = \frac{\alpha^2 E_s}{N_0}$, and the average SNR per symbol by $\overline{\gamma} = \Omega E_s / N_o$, where $E_s$ is the energy per symbol [12].

Depending on the nature of the radio propagation environment, there are different models describing the statistical behavior of the multipath fading envelope. The Rayleigh distribution is frequently used to model multipath fading with no direct line-of-sight (LOS) path. In this case, the channel fading amplitude is distributed according to Rayleigh distribution function and it is given by

$$f_\alpha\left(\alpha\right) = \frac{\alpha}{\Omega} e^{\frac{-\alpha^2}{2\Omega}}. \tag{3}$$

Likewise, the instantaneous SNR per symbol $\gamma$ of the channel is distributed according to an exponential distribution given by [12].

$$f_\gamma\left(\gamma\right) = \frac{1}{\overline{\gamma}} e^{\frac{\gamma}{\overline{\gamma}}}. \tag{4}$$

According to Equation (1), the MGF corresponding to this fading model is given by

$$M_\gamma\left(s\right) = \left[1 - \overline{\gamma}s\right]^{-1}. \tag{5}$$

## 3. BER Considering Leaky Feeder Transmitter

The LF has uniformly or randomly distributed transversal slots. The feeder's linear axial attenuation is assumed to be $k$ dB/m at a certain frequency $f$, with the relative phase velocity of $\beta$. Each transverse LF slot can be approximated by the radiation profile of a dipole antenna in the Rayleigh fading channel [10,14]. Hence, the LF can be modeled as an array group of radiating dipoles lying on the $y = 0$ axis of a two-dimensional plane. Starting from $x = 0$, we consider $N$ slots at coordinates $(D(n), 0)$ for $n = 1, 2, ..., N$, as shown in Figure 1.

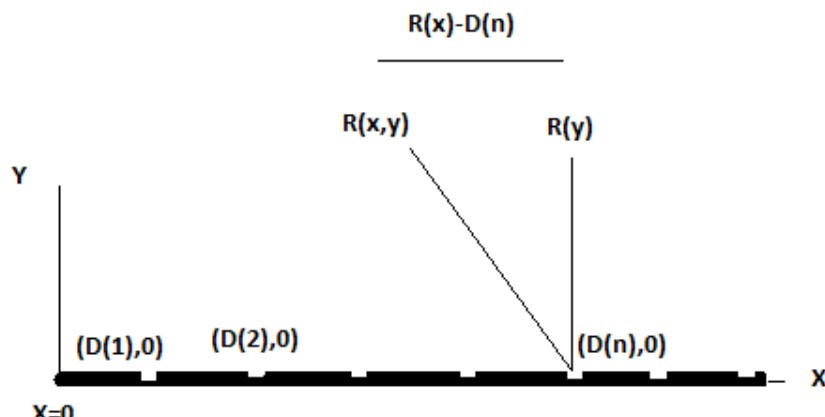

**Figure 1.** Basic configuration of the leaky coaxial cable.

*3.1. Calculation of BER of a QPSK Modulation Technique*

The attenuated and phase-rotated MPSK signal emanating from the $n^{th}$ slot can be described as in (6) [9].

$$S(t) = \sqrt{\left(\frac{2\,E_b}{T}\right)} \cos\left(2\,\pi\,f_c\,t + \frac{(2\,m-1)\,\pi}{4}\right) 10^{\frac{-D\,n\,k}{20}} e^{\frac{j2\,\pi\,\beta\,D\,n}{\lambda}}, \tag{6}$$

where $m = 1, 2, 3, \ldots M$ is the constellation number of the M-phase shift keying; $\lambda$ is the wavelength and $E_b$ is the energy per bit. For a single slot, $E_b = \frac{A^2\,T_b}{2}$, where $A$, $T_b$ are the amplitude of the carrier and bit duration, respectively.

Since the signal emitted from each slot has noise added and travels along a Rayleigh fading channel, it can be modeled as a MISO system as shown in Figure 2.

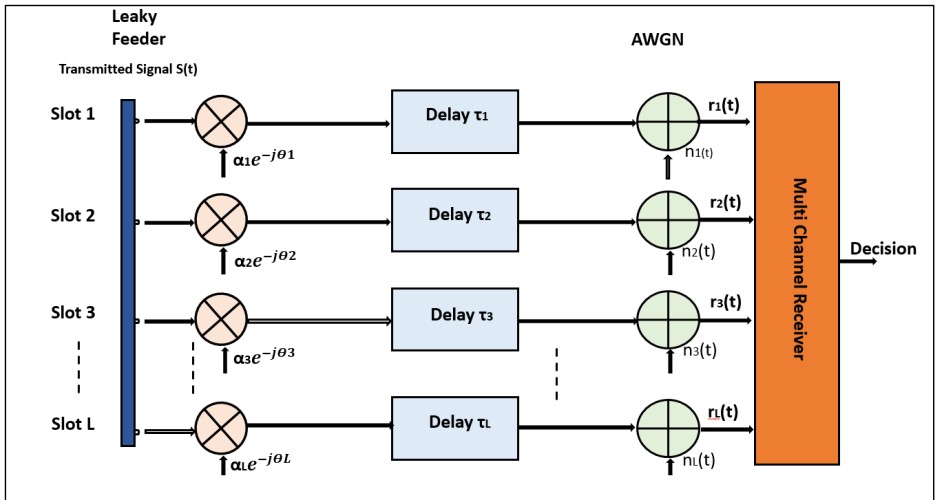

**Figure 2.** Leaky feeder channel model.

Since the received signal in a multipath channel consists of a series of attenuated, time-delayed, and phase-shifted replicas of the transmitted signal, the channel can be modeled as shown in Figure 3. which is a complex impulse response in the time domain.

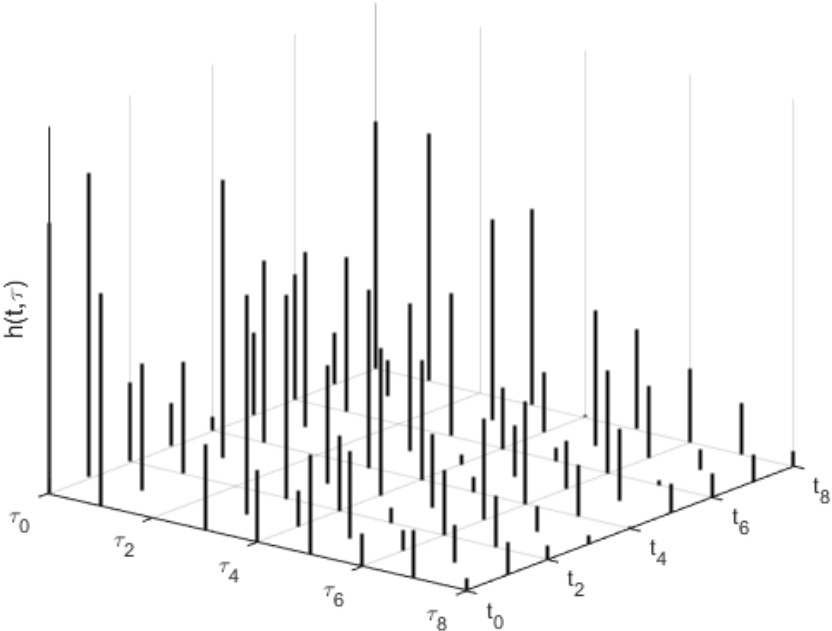

**Figure 3.** Simulation for a mathematical model of the multipath impulse response.

The base band impulse response can be written as [15]

$$h\left(t,\tau\right) = \sum_{l=1}^{L} a_l\left(t\right) e^{j\theta_l(t)} \delta\left(\tau - \tau_l\left(t\right)\right), \tag{7}$$

where $a_l\left(t\right)$, $\tau_l\left(t\right)$, and $\theta_l\left(t\right)$ are random variables expressing indicate the amplitude, excess delay, and phase shift of multiple paths, respective to the reference path $l$. In addition, $L$ is the number of multipath components at the measurement instant. $\delta(.)$ represents the Dirac delta function.

The received signal is given by

$$y\left(t\right) = \sum_{n=1}^{N} \sum_{l=1}^{L} S^*\left(t\right) a_l\left(t\right) e^{j\theta_l(t)} \delta\left(t - \tau_l\left(t\right)\right), \tag{8}$$

where $N$ is the number of equally spaced slots and $*$ is the convolution sign.

The theoretical BER of the QPSK modulation under an AWGN channel can be given by the following standard expression:

$$P_b = Q\left(\sqrt{\left(\frac{2E_b}{N_0}\right)}\right). \tag{9}$$

Similarly, the theoretical BER of the QPSK modulation over a Rayleigh fading channel for a single antenna is given by [12]

$$P\left(E\right) = \frac{3}{4}\left\{\left(\left(1 - \sqrt{\left(\frac{0.5\,\overline{\gamma_t}}{1 + 0.5\,\overline{\gamma_t}}\right)}\right)\left(\frac{4}{3\,\pi}\right)\right)\left(\left(\frac{\pi}{2}\right) + \arctan\left(\sqrt{\left(\frac{0.5\,\overline{\gamma_t}}{1 + 0.5\,\overline{\gamma_t}}\right)}\cot\left(\frac{\pi}{4}\right)\right)\right)\right\}. \tag{10}$$

For the LF cable, the signal that will be transmitted form an arbitrary slot is shown in (6); this equation can be rewritten in a phaser form as

$$S(t) = A e^{j\omega_c} e^{\frac{j(2m-1)\pi}{4}} 10^{\frac{-Dnk}{20}} e^{\frac{j2\pi Dn\beta}{\lambda}}. \tag{11}$$

If the slots are close enough so that the relative phase rotation and time delay can be ignored($e^{\frac{j2\pi nD\beta}{\lambda}} \cong 1$) (but not the fading), the sum of the signals will be given by the real part of

$$S(t)_T = \sum_{n=1}^{N} A e^{j\omega_c} e^{\frac{j(2m-1)\pi}{4}} 10^{\frac{-Dnk}{20}} e^{\frac{j2\pi Dn\beta}{\lambda}}. \tag{12}$$

In this case, the new amplitude of the sum of the signals will be

$$A_T = \sqrt{\left(\left(\sum_{n=1}^{N} A\, 10^{\frac{-Dnk}{20}} \cos\left(\frac{(2m-1)\pi}{4} + \frac{2\pi Dn B}{\lambda}\right)\right)^2 + \left(\sum_{n=1}^{N} A\, 10^{\frac{-Dnk}{20}} \sin\left(\frac{(2m-1)\pi}{4} + \frac{2\pi Dn B}{\lambda}\right)\right)^2\right)}. \tag{13}$$

The bit energy of the received signal is given by

$$E_b = \frac{A^2_T T_b}{2}, \tag{14}$$

and the SNR of the $l^{th}$ path under Rayleigh fading is given by

$$\gamma_l = \frac{E_b}{N_0} \alpha^2{}_l. \tag{15}$$

The total SNR from all paths $\gamma_t$ will be given by [13]

$$\gamma_t = \sum_{l=1}^{L} \gamma_l. \tag{16}$$

Substituting Equation (15) in Equation (16), the $\gamma_t$ can be in the form of

$$\gamma_t = \sum_{l=1}^{L} \frac{A^2_T T_b}{N_0} \alpha^2{}_l. \tag{17}$$

By substituting (13) in (17), the total SNR $\gamma_t$ is given by

$$\gamma_t = \sum_{l=1}^{L} \left\{ \left[\sum_{n=1}^{N} A\, 10^{\frac{-Dnk}{20}} \cos\left(\frac{(2m-1)\pi}{4} + \frac{2\pi Dn\beta}{\lambda}\right)\right]^2 \right.$$
$$\left. + \left[\sum_{l=1}^{N} A\, 10^{\frac{-Dnk}{20}} \sin\left(\frac{(2m-1)\pi}{4} + \frac{2\pi Dn\beta}{\lambda}\right)\right]^2 \right\} \cdot \left\{\frac{T_b \alpha^2_l}{N_0}\right\}. \tag{18}$$

Define the conditional probability of error of multipath as [16],

$$P_b\left(E \setminus \{\gamma_l\}_{l=1}{}^L\right) = \frac{1}{\pi} \int_0^{\left(\frac{3*\pi}{4}\right)} e^{\frac{-g\gamma_t}{\sin^2(\phi)}} \, d\phi, \tag{19}$$

where $g = sin^2\left(\frac{\pi}{M}\right)$, and $M$ is the constellation number ($M = 4$ for QPSK modulation).

If all paths (Gammas) are assumed to be independent, then the probability density function of the combined SNR $\gamma_t$ has a chi-Square distribution [12], and can be written as

$$f_{\gamma t}\left(\gamma_t\right) = \frac{1}{(L-1)!\,\overline{\gamma}^L}\,(\gamma_t)^{L-1}\,e^{\frac{-\gamma_t}{\overline{\gamma}}}. \tag{20}$$

According to (17), the average SNR $\overline{\gamma}$ is given by

$$\overline{\gamma_t} = \sum_{l=1}^{L}\left\{\left[\sum_{n=1}^{N} A\,10^{\frac{-Dnk}{20}}\,\cos\left(\frac{(2m-1)\,\pi}{4} + \frac{2\,\pi\,D\,n\,\beta}{\lambda}\right)\right]^2\right.$$
$$\left. + \left[\sum_{l=1}^{N} A\,10^{\frac{-Dnk}{20}}\,\sin\left(\frac{(2m-1)\,\pi}{4} + \frac{2\,\pi\,D\,n\,\beta}{\lambda}\right)\right]^2\right\}\cdot\left\{\frac{T_b\,E[\alpha_l^2]}{N_0}\right\}. \tag{21}$$

By substituting (19) and (20) in (2), for QPSK when $M = 4$, the BER of the LF will be given as

$$P_b\left(E\right) = \frac{1}{\pi}\int_0^{\left(\frac{3\pi}{4}\right)}\prod_{l=1}^{L} M_{\gamma_l}\left(\frac{-g\,\gamma_l}{sin^2\left(\phi\right)}\right)\,d\phi, \tag{22}$$

where the closed form of (22) will be [13],

$$P_b\left(E\right) = \left(\frac{1-\mu}{2}\right)^L\sum_{l=0}^{L-1}\binom{L-1+l}{l}\left(\frac{1+\mu}{2}\right)^l. \tag{23}$$

Here, $\mu = \sqrt{\left(\frac{\overline{\gamma}}{1+\overline{\gamma}}\right)}$.

To show the effect of the LF parameters on the BER, substitute (21) in (23):

$$P_b\left(E\right) = I_1\,I_2, \tag{24}$$

where $I_1$ and $I_2$ are given by (25) and (26), respectively, and $\overline{\gamma_t}$ is given by (21):

$$I_1 = \left(0.5 - 0.5\sqrt{\left(\frac{\overline{\gamma_t}}{1+\overline{\gamma_t}}\right)}\right)^L, \tag{25}$$

$$I_2 = \sum_{1=l}^{L}\binom{L+l-1}{l}\left(0.5 + 0.5\sqrt{\left(\frac{\overline{\gamma_t}}{1+\overline{\gamma_t}}\right)}\right)^l. \tag{26}$$

If $\overline{\gamma_t} >> 1$, then (23) can be simplified to

$$P_e\left(E\right) = \frac{\binom{2L-1}{L}}{\left(4\,\overline{\gamma_t}\right)^L}. \tag{27}$$

Equation (27) shows clearly that the BER of the LF depends on inter slot distance $D$, the attenuation of the cable $k$ and on the number of slots $N$ in addition to the path fading gain $\alpha_l$.

### 3.2. BER for M-QAM Modulation Technique

M-ary Quadrature Amplitude Modulation (QAM) is a widely used modulation technique that provides high transmission rates with high bandwidth efficiency and with the correct configuration,

high energy efficiency. However, finding the expression of the bit error probability of M-ary QAM is not as straightforward as finding its symbol error probability. Square M-ary QAM involves the amplitude modulation of two carriers in the quadrature expressed as [17]

$$S(t) = \sqrt{\left(\frac{2E}{T}\right)} A_x \cos\left(2\pi f_c t\right) - \sqrt{\left(\frac{2E}{T}\right)} B_x \sin\left(2\pi f_c t\right), \tag{28}$$

where $A_x$ and $B_x$ are the signal amplitudes of the in-phase and quadrature components, respectively, $T$ is the symbol duration, $f_c$ is the carrier frequency, $1 < x < M$, $M$ is number of constellation, and $M = 2^k$, where k is an even.

For the LF, the transmitted signal for $n^{th}$ slot will be given by

$$S_n(t) = \left\{\sqrt{\left(\frac{2E}{T}\right)} A_x \cos\left(2\pi f_c t\right) - \sqrt{\left(\frac{2E}{T}\right)} B_x \sin\left(2\pi f_c t\right)\right\} \cdot 10^{\frac{-Dnk}{20}} e^{\frac{j2\pi Dn\beta}{\lambda}}. \tag{29}$$

The total signal which is a sum of all slots signals is given by

$$S_T(t) = \sum_{n=1}^{N} \left\{\sqrt{\left(\frac{2E}{T}\right)} A_x \cos\left(2\pi f_c t\right) - \sqrt{\left(\frac{2E}{T}\right)} B_x \sin\left(2\pi f_c t\right)\right\} \cdot 10^{\frac{-Dnk}{20}} e^{\frac{j2\pi Dn\beta}{\lambda}}. \tag{30}$$

Assume $A_x = B_x = C_x$, and use the following identity

$$\cos(z) - \sin(z) = \sqrt{(2)} \sin\left(\frac{\pi}{4} - z\right). \tag{31}$$

The total signal will be written as

$$S_T(t) = \sum_{n=1}^{N} \left\{2\sqrt{\left(\frac{E}{T}\right)} C_x \sin\left(\frac{\pi}{4} - 2\pi f_c t\right) 10^{\frac{-Dnk}{20}} e^{\frac{j2\pi Dn\beta}{\lambda}}\right\}. \tag{32}$$

By using Euler's identity, $S_T(t)$ can be written as

$$S_T(t) = \sum_{n=1}^{N} \left\{2\sqrt{\left(\frac{NE}{T}\right)} C_x \cos\left(2\pi f_c t + \frac{\pi}{4} + \arctan\left(\frac{2\pi n D\beta}{\lambda}\right)\right)\right\}. \tag{33}$$

Then, the total amplitude of all signals transmitted from all slots $A_T$ will be as

$$A_T = \sum_{n=1}^{N} \left(2\sqrt{\left(\frac{E_b N}{T}\right)} C_x \left(10^{\frac{-Dnk}{20}}\right)\right). \tag{34}$$

To calculate the BER of the LF which carries M-QAM signal, we have to find the conditional probability first, which is given by [13]

$$P\left(E \setminus \{\gamma_l\}_{l=1}^{L}\right) = 4\left(1 - \frac{1}{\sqrt{(M)}}\right) Q\left(\sqrt{(2 g_{QAM} \gamma_t)}\right)$$

$$- 4\left(1 - \frac{1}{\sqrt{(M)}}\right)^2 Q^2\left(\sqrt{(2 g_{QAM} \gamma_t)}\right), \tag{35}$$

where $g_{QAM} = \frac{3}{2M-2}$, Simon and Divsalar in [18], generalized the alternative representation of the Gaussian $Q$ function to the two-dimensional case and showed in particular

$$Q(x) = \frac{1}{\pi} \int_0^{\left(\frac{\pi}{2}\right)} \exp\left(\frac{-x^2}{2\sin^2(\phi)}\right) d\phi \tag{36}$$

and

$$Q^2(x) = \frac{1}{\pi} \int_0^{\left(\frac{\pi}{4}\right)} \exp\left(\frac{-x^2}{2\sin^2(\phi)}\right) d\phi \tag{37}$$

so the conditional probability $P\left(E \setminus \{\gamma_l\}_{l=1}{}^L\right)$ under the AWGN channel can be written as

$$P\left(E \setminus \{\gamma_l\}_{l=1}{}^L\right) = \frac{4}{\pi}\left(1 - \frac{1}{\sqrt{(M)}}\right) \int_0^{\left(\frac{\pi}{2}\right)} \exp\left(\frac{-g_{QAM}\,\gamma_t}{\sin^2(\phi)}\right) d\phi$$
$$-\frac{4}{\pi}\left(1 - \frac{1}{\sqrt{(M)}}\right)^2 \int_0^{\left(\frac{\pi}{4}\right)} \exp\left(\frac{-g_{QAM}\,\gamma_t}{\sin^2(\phi)}\right) d\phi. \tag{38}$$

The BER for a single antenna transmitted a M-QAM signal in Rayleigh fading channel is given by [13]

$$P(E) = 2\left(1 - \frac{1}{\sqrt{(M)}}\right)\left(1 - \sqrt{\left(\frac{1.5\,\gamma_t}{M - 1 + 1.5\,\gamma_t}\right)}\right)$$
$$-\left(\left(1 - \frac{1}{\sqrt{(M)}}\right)^2\right)\left(1 - \sqrt{\left(\frac{1.5\,\gamma_t}{M - 1 + 1.5\,\gamma_t}\right)}\right)$$
$$\left(\left(\frac{4}{\pi}\right)\arctan\left(\frac{\sqrt{(M - 1 + 1.5\,\gamma_t)}}{1.5\,\gamma_t}\right)\right). \tag{39}$$

By using (34), and (17), $\gamma_t$ is given by

$$\gamma_t = \sum_{l=1}^L \frac{\left(\sum_{n=1}^N 2\sqrt{\left(\frac{E_b\,N}{T_b}\right)}\,C_x \cdot 10^{\frac{-Dnk}{20}}\right)^2 T_b \cdot \alpha_l^2}{N_0} \tag{40}$$

and $\overline{\gamma_t}$ is given by

$$\overline{\gamma_t} = \sum_{l=1}^L \frac{\left(\sum_{n=1}^N 2\sqrt{\left(\frac{E_b\,N}{T_b}\right)}\,C_x \cdot 10^{\frac{-Dnk}{20}}\right)^2 T_b\,E\left[\alpha_l^2\right]}{N_0}. \tag{41}$$

For MRC with $L$ independent identical channels (i.i.d), the BER for M-QAM can be obtained by averaging (38) over (20) as in (2)

$$P(E) = \left\{\int_0^\infty \frac{1}{(L-1)!\,\overline{\gamma_t}}\,\gamma_t^{L-1}\,e^{\frac{\gamma_t}{\overline{\gamma_t}}}\,d\gamma_t \left\{\frac{4}{\pi}\left(1 - \frac{1}{\sqrt{(M)}}\right)\int_0^{\left(\frac{\pi}{2}\right)} \exp\left(\frac{-g_{QAM}\,\gamma_t}{\sin^2(\phi)}\right) d\phi\right.\right.$$
$$\left.\left. -\frac{4}{\pi}\left(1 - \frac{1}{\sqrt{(M)}}\right)^2 \int_0^{\left(\frac{\pi}{4}\right)} \exp\left(\frac{-g_{QAM}\,\gamma_t}{\sin^2(\phi)}\right) d\phi\right\}\right\}. \tag{42}$$

The integral in (42) can be divided into two integrals such as

$$P(E) = I_1 - I_2, \tag{43}$$

where $I_1$ and $I_2$ is given by

$$I_1 = \left\{ \int_0^\infty \frac{1}{(L-1)!\,\overline{\gamma_t}} \gamma_t^{L-1} e^{\frac{\gamma_t}{\overline{\gamma_t}}} \, d\gamma_t \left\{ \frac{4}{\pi} \left( 1 - \frac{1}{\sqrt{(M)}} \right) \int_0^{\left(\frac{\pi}{2}\right)} \exp \left( \frac{-g_{QAM}\,\gamma_t}{\sin^2(\phi)} \right) d\phi \right\} \right\}, \tag{44}$$

$$I_2 = \left\{ \int_0^\infty \frac{1}{(L-1)!\,\overline{\gamma_t}} \gamma_t^{L-1} e^{\frac{\gamma_t}{\overline{\gamma_t}}} \, d\gamma_t \left\{ \frac{4}{\pi} \left( 1 - \frac{1}{\sqrt{(M)}} \right)^2 \int_0^{\left(\frac{\pi}{4}\right)} \exp \left( \frac{-g_{QAM}\,\gamma_t}{\sin^2(\phi)} \right) d\phi \right\} \right\}. \tag{45}$$

According to [19,20], $I_1$ and $I_2$ can be evaluated as

$$I_1 = q \left( \frac{1-\mu}{2} \right)^L \sum_{l=0}^{L-1} \binom{L-1+l}{l} \left( \frac{1+\mu}{2} \right)^l, \tag{46}$$

where

$$q = 4 \left( 1 - \frac{1}{\sqrt{(M)}} \right) \tag{47}$$

and

$$\mu = \sqrt{\left( \frac{\overline{\gamma_t}}{p + \overline{\gamma_t}} \right)} \tag{48}$$

and

$$p = \frac{2(M-1)}{3\,log_2\,M}. \tag{49}$$

$I_2$ given by

$$I_2 = \left( \frac{q^2\,p^L}{16 \cdot (L-1)! \cdot \gamma_t^L} \right) \left\{ \frac{(L-1)!}{a^L} - \left( \frac{4}{\sqrt{(\pi)}\,a^L} \sum_{l=1}^L \frac{(-1)^{l-1} \cdot (L-1)!\,a^L\,I_3}{l!} \right) \right\}, \tag{50}$$

where

$$a = \frac{p}{\gamma_t}, \tag{51}$$

$$b = \frac{p + \overline{\gamma_t}}{\gamma_t}, \tag{52}$$

$$I_3 = \frac{l!}{(2l-1)\,\sqrt{(\pi)}} \, {}_2F_1\,(0.5+l, 1+l; 1.5+l; -b), \tag{53}$$

where ${}_2F_1\,(0.5+l, 1+l; 1.5+l; -b)$ is the hypergeometric function. Therefore, combining (46)–(53), and subsituting the results in (43) gives the desired BER.

### 4. Simulation and Numerical Results

Simulation was done using Matlab software R2017b to verify the performance of the LF MISO system. We have considered QPSK modulation and a Rayleigh flat fading channel and perfect phase matching between the transmitter and the receiver. The fixed parameters used in the simulation are shown in Table 1. The values of the variable parameters such as the slot period $D$ and the carrier frequency $f$ (i.e., $\lambda$) are shown in each figure as appropriate.

**Table 1.** Parameters used in the simulation processes.

| | |
|---|---|
| Carrier Amplitude $A$ | 15 V |
| Phase velocity $\beta$ | 0.9 rad/s |
| Number of Paths $L$ | 3 |
| Noise Power $N_0$ | 0 dB |
| Axial attenuation $k$ | 0.15 dB/km |
| Bit Duration $T_b$ | 0.1 ms |
| Leaky Feeder's length | 100 m |

Figure 4 is generated at carrier frequency 3 GHz. It can be seen that the BER has a periodic fluctuation because of the phase rotating nature of the LF. The lowest BER occurs at a slot period (separation) of 20 mm. Probably, this is the optimal value for the given frequency. However, note that the slot separation has to be practically feasible for a given LF cable (based on its length, loss requirement and other factors). As the ratio between the transmitted and conducted RF energy depends on the slot period, too small separation will cause too much energy to be radiated and vice versa. The general trend of increment in the BER as the slot period increases happens because the amount of energy received by the receiver decreases with high slot spacing. This is probably because of a lesser number of (radiating) slots in a given length (100 m). Essentially, this figure shows that, by appropriately optimizing the slot period of an LF for a given carrier frequency, a very low BER can be achieved.

Figures 5 and 6 show a very important outcome. *The leaky feeder provides a significantly lower BER compared to a single antenna for the same SNR* in both QPSK and M-QAM modulation schemes. It is important to interpret these graphs carefully. In these figures, the *SNR* of both systems are equal, not the transmission power. The LF has to be fed with more power to achieve the same SNR because of the multiple slots. Nevertheless, the LF system, acting as a distributed transmitter, will have lower RF power density and will cause less (co-channel) interference with neighboring systems while offering much lower BER. This is an interesting observation.

Figure 7 is plotted in order to study the effect of RF carrier frequency on BER. Here, the $x$-axis is the wavelength ($\lambda$) of the RF signal as a function of the slot period $D$, and $D$ is fixed at 200 mm. This figure shows that the minimum BER will be achieved whenever $\lambda = D/I$, where $I$ is an integer for $\lambda \leq D$. This is because, at this wavelength, *constructive interference* occurs. Since $D = 200$ mm, the corresponding frequency is $0.3 < f < 30$ GHz (i.e., $0.1 < \lambda < 1$ m). We note that the BER also shows a periodic oscillation when $\lambda < D$ and then steadily decreases for large values of $\lambda$. Since BER depends on the slot distance $D$ as well, Figures 4 and 6 have to be jointly used to study the behavior of the LF system.

Figure 8 shows the BER of the LF for different values of M in the M-QAM modulation scheme. Note that $M = 4$ gives the best BER, as higher values of $M$ are affected by larger phase errors, due to the phase rotation in the LF itself.

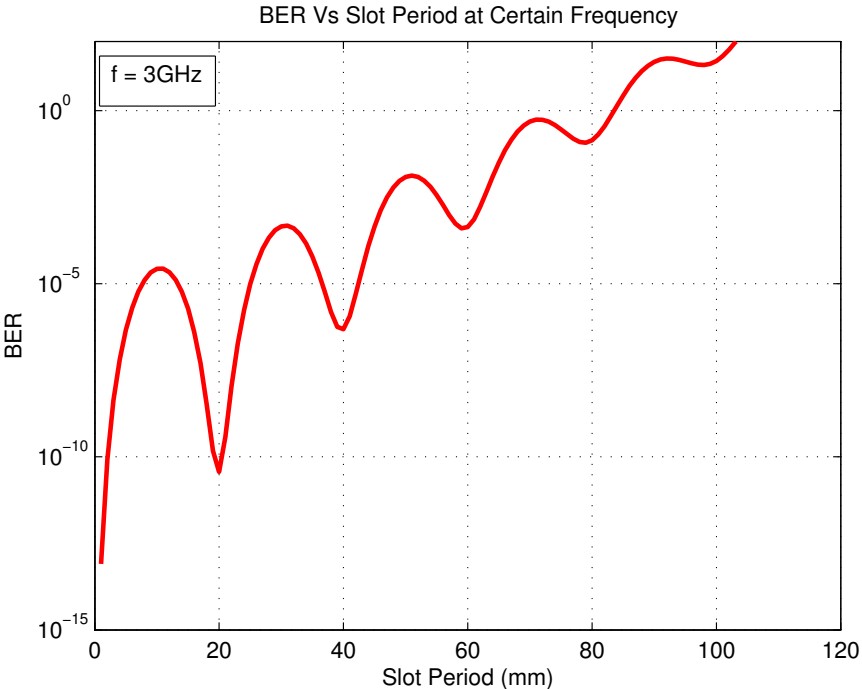

**Figure 4.** Bit Error Rate vs. the slot period (number of slots) of the leaky feeder.

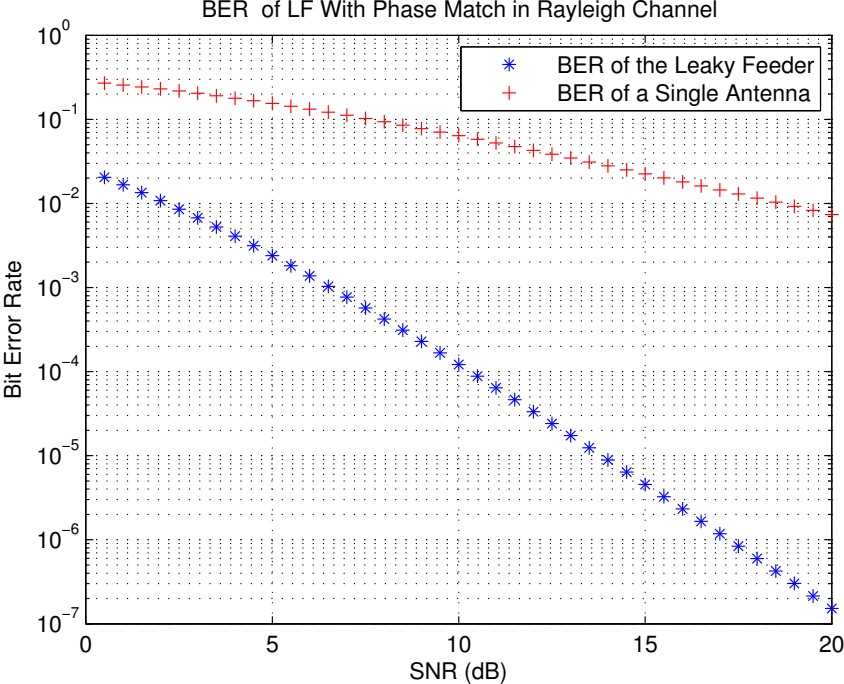

**Figure 5.** BER of Leaky Feeder with phase matching.

The received power versus the distance is illustrated in Figure 9. Note that the received power was acquired by simulating the LF and a single antenna using Matlab software by assuming the receiver distance up to 50 m. The result shows a stability in the received power when LF is in use, where the received power level is varying from [–15 to –25] (dBm), while, when the single antenna is in use, the received power level is varying from [–5 to –50] (dBm). The explanation for the superiority of LF to single antenna is as follows: the LF is acting as a MIMO system where the received signal

consists of many components coming from different slots. Note that the power received at the user would be the sum of powers emitted from multiple slots. This is typically better than many single antenna systems.

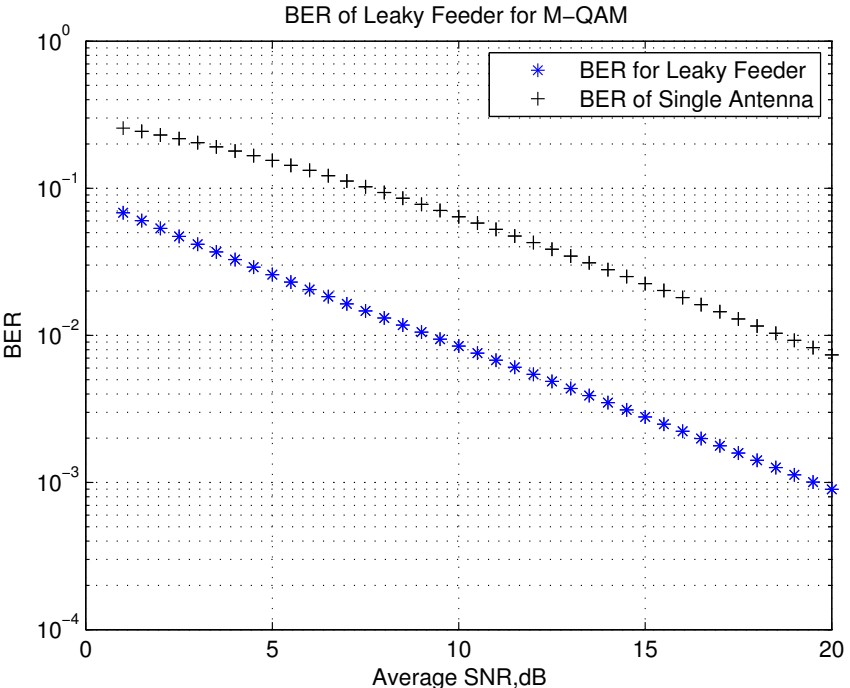

**Figure 6.** BER of LF vs. single antenna.

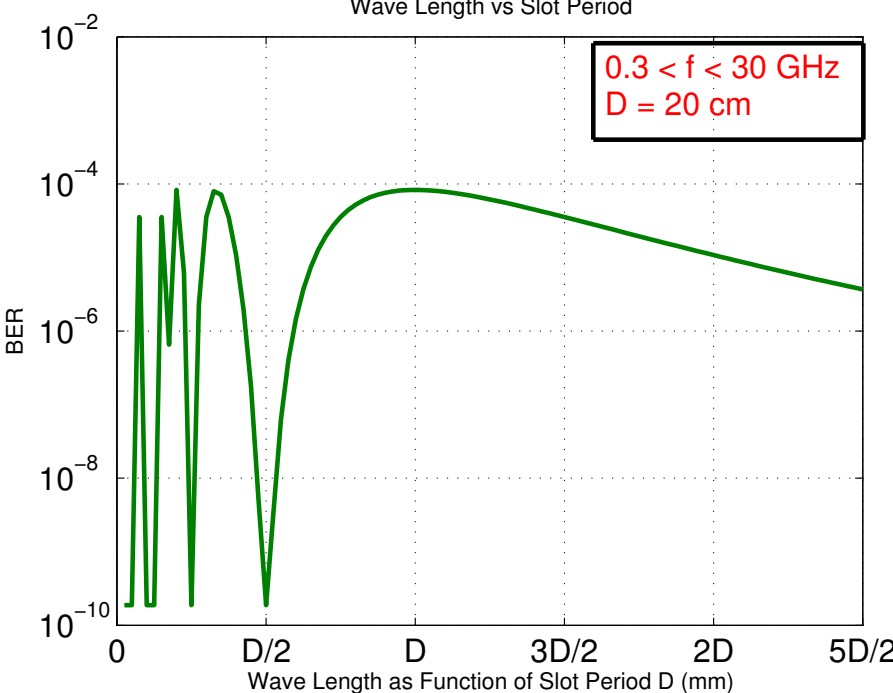

**Figure 7.** BER of LF with the wave length.

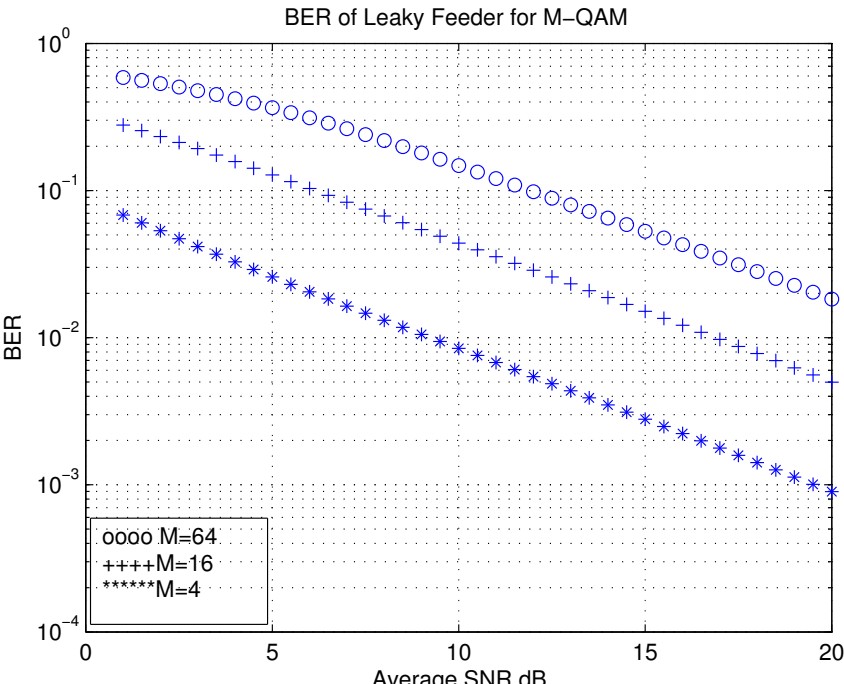

**Figure 8.** BER of LF for different values of M.

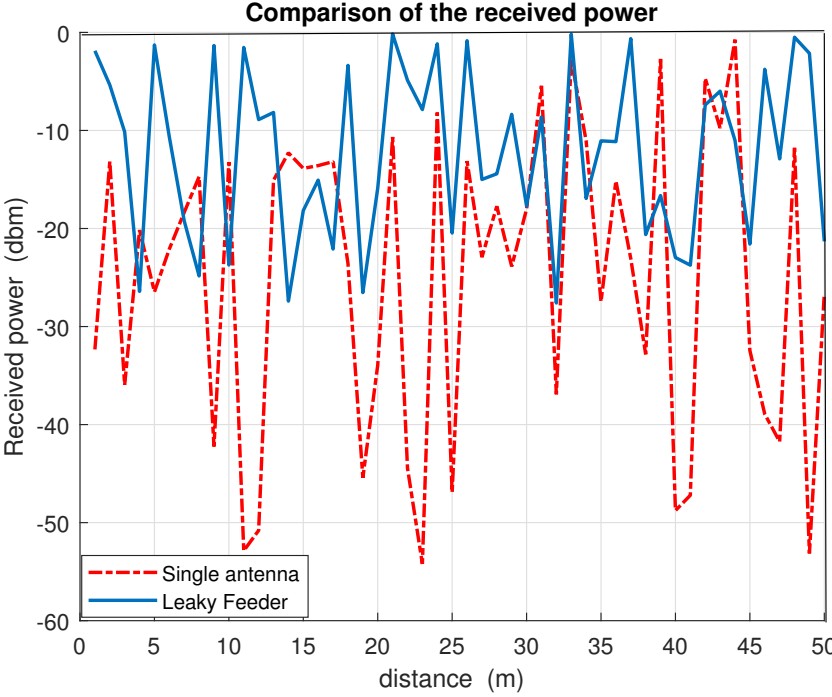

**Figure 9.** Comparison of the received power.

## 5. Conclusions and Future Work

In this paper, the leaky feeder is modeled as a (large) multi-antenna array transmitter and the BER performance is evaluated under a Rayleigh flat fading channel. Analytical expressions are derived and simulation is performed to understand the system behavior with respect to variation in a number of parameters such as the slot spacing, radio frequency, and modulation scheme.

Our observations tell that the leaky feeder MISO system can yield a satisfactory solution to V2I communication problems in tunnels, better than a single antenna transmitter system for the same receiver SNR. However, the system has to be optimized in terms of slot spacing for a given radio frequency to achieve the best BER. The slot spacing also affects both the BER and the loss along the cable (hence, amplifier free length of the cable).

The LF also has an added benefit of easy installation and low cost. Since both the *transmission and radiation* functionalities are carried out by just one cable, the cost and complexities associated with having multiple antennas and their matching/driving circuits are alleviated.

Studying the behavior of LF under wide band conditions would be an interesting task. Moreover, studying the system under a more realistic Rician fading channel will be considered next.

**Author Contributions:** All authors have equal contribution.

**Funding:** This research received no external funding.

**Conflicts of Interest:** The authors declare no conflict of interest.

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
