# Peer review of "The Leaky Feeder, a Reliable Medium for Vehicle to Infrastructure Communications"

_asi, doi:10.3390/asi2040036_

Round 1

Reviewer 1 Report

This work is very interesting with a solid methodology. However, there are some points that should be addressed

1) Although the mathematical context is adequate, the analysis should focus, also, to the extracted results since the BER comparison is the main subject.

2) One of the main results of this paper is that the leaky feeder antenna presents enhanced BER compared to the single element one for equal SNR. However, this observation is expected and the power issues of the leaky feeder have to be addressed. Specifically, it is written in line 165 that more power is required and this is a critical point (especially in M-QAM) that the authors are not discussing adequately. Maybe a study towards the energy consumption (as well as the losses of the cable) should be considered.

3) The authors claim in conclusion that the theoretically extracted analytical expressions are numerically verified but there is not any verification in the main document.

4) Most of the figures are not close to their reference. For example Figure 4 is at page 7 and it is called at page 11. This fact makes reading very difficult

Author Response

"Please see the attachment

Reviewer 2 Report

Although the paper has some merits, I have few general concerns that need to be addressed carefully before I can recommend for publications:

(1) Figures 2 and 3 are not clear. We think they are borrowed from other papers/books. Authors need to redraw the figures.

(2) The parameter table should come before the figures.

(3) It is strange that the paper is about V2X communication, but no V2X paper is cited in the paper. We strongly suggest to include following recent papers while discussing the proposed technique's applicability for DSRC and LTE V2X network.

(i) Nguyen et al "semi-empirical performance study of two-hop DSRC message relaying at road intersections" MDPI Information 2018.

(ii) Jianhua He 'Enhanced Collision Avoidance for Distributed LTE Vehicle to Vehicle Broadcast Communications', 2018

(iii) Noor, "Broadcast Performance Analysis and Improvements of the LTE-V2V Autonomous Mode at Road Intersection," IEEE TVT, 2019.

(iv) Francisco et al 'Geolocation-Based Access for Vehicular Communications: Analysis and Optimization via Stochastic Geometry', 2018.

(4) Please rewrite the motivation of the proposed work clearly.

Round 2

Reviewer 1 Report

The authors addressed the majority of the recommendations and the paper is significantly improved. However, there is one more point relative to the power consideration:

The finding that 400mW is enough for adequate performance is a bit arbitrary since the receiver gain and the required power are not considered and most importantly are not cited. Moreover, there is not any comparison to the case of a single element case that possibly the losses are significantly lower. This should be addressed to indicate the importance of the proposed leaky feeder.

Reviewer 2 Report

It is very unfortunate that the authors made a rush revision based on our comments. We ask the authors again to carefully revise the paper according to the comments. Specifically:

(i) The figures are not still in good quality.

(ii) We suggested to include V2X related paper. The authors claim that they have added "many relevant reference", while ended up just adding 3 relevant works and   ignored one of the suggested works. What the authors should do is include all the suggested papers along with some other relevant V2X paper.

(iii) The changes in the paper should be highlighted, otherwise it is hard to follow where the changes have been made.

Round 3

Reviewer 2 Report

We are satisfied with the revised version.